# Tocilizumab Evaluation in HLA-Desensitization before Kidney Transplantation as an Add-On Therapy to Apheresis: The TETRA Study

**DOI:** 10.3390/jcm12020424

**Published:** 2023-01-04

**Authors:** Thomas Jouve, Mélanie Daligault, Johan Noble, Florian Terrec, Farida Imerzoukene, Céline Dard, Béatrice Bardy, Paolo Malvezzi, Lionel Rostaing

**Affiliations:** 1Service de Néphrologie, Hémodialyse, Aphérèses et Transplantation Rénale, CHU Grenoble-Alpes, 38043 Grenoble, France; 2Institute for Advanced Biosciences (IAB), INSERM U 1209, CNRS UMR 5309, Université Grenoble Alpes, 38400 Grenoble, France; 3Faculty of Health, Université Grenoble Alpes, 38400 Saint-Martin-d’Hères, France; 4Etablissement Français du Sang (EFS) Rhône Alpes, 38700 La Tronche, France

**Keywords:** desensitization, kidney transplantation, tocilizumab, anti-HLA alloantibodies, MFI

## Abstract

Background: Desensitization strategies improve access to transplantation in highly sensitized kidney transplant candidates. Tocilizumab could be a valuable addition to more traditional desensitization regimens. We investigated the effect of tocilizumab as an add-on therapy to our standard of care (SoC) desensitization strategy based on rituximab and apheresis. Methods: In this study, we prospectively included highly sensitized patients to receive monthly tocilizumab infusions for 6 months before our SoC regimen (Toci + SoC group). We compared the reductions in the mean fluorescent intensity (MFI) rebound at post-transplantation and kidney function at 1-year post-transplantation to patients treated by SoC (based on apheresis and two doses of rituximab). Results: Twenty-six patients were included in the SoC group; seven in the Toci + SoC group. Reductions in pre-transplantation MFI were similar between groups. At 1-year post-transplantation, there was no absolute difference in overall MFI rebounds, including donor-specific antibodies. Toci + SoC helped lower the rebound of antibodies with more elevated baseline MFIs. Graft function and survival rates were similar at one-year post-transplantation (median eGFR 62.8 vs. 65.6 mL/min/1.73 m^2^ for SoC and Toci + SoC, respectively). Conclusions: Tocilizumab as an add-on to SoC desensitization may help control the post-transplantation rebound of antibodies with elevated baseline MFIs. However, reductions in pre-transplantation MFIs were similar with or without tocilizumab. Further studies are needed to validate this pilot study.

## 1. Introduction

The number of highly allo-sensitized patients is increasing amongst kidney transplant candidates. Access to transplantation remains a challenge for these patients despite improvements in desensitization strategies. These strategies rely on apheresis to remove HLA allo-antibodies, and B-cell targeting drugs (mainly rituximab) to limit or prevent reconstitution of these allo-antibodies. Furthermore, these strategies must control or mitigate the allo-antibody rebound after transplantation to prevent or limit antibody-mediated rejection (ABMR). Reassuring results for such HLA desensitization strategies have been provided in recent years with various desensitization protocols, including rituximab, IV immunoglobulins (IVIg), and various apheresis techniques [1,2,3,4].

More recently, interleukin (IL)-6-targeting drugs have been explored for their potential impact on humoral immunity [5,6,7]. They target follicular helper T lymphocytes and plasma cells with the goal of sustainably depleting allo-antibody-producing cells. These pilot studies have shown some efficiency but have not been designed to assess the added value of IL-6 targeting in addition to the usual desensitization protocols (based mainly on various combinations of rituximab, IVIg, and apheresis).

We have recently shown that tocilizumab alone, as a desensitization therapy, has a very limited impact on HLA antibodies [8]. Access to transplantation is not made easier using tocilizumab alone. However, tocilizumab induces a defect in B cell maturation [9]. This suggests a potential role for tocilizumab to prevent HLA-antibody rebound after transplantation.

In this single-center non-randomized prospective study, we evaluated the effect of tocilizumab as an add-on therapy to our standard of care (SoC) desensitization regimen, both in terms of reducing HLA antibodies, as assessed by mean fluorescent intensity (MFI), and 1-year immunological and functional outcomes.

## 2. Materials and Methods

### 2.1. Patients

We prospectively included kidney transplant candidates who had elevated levels of calculated panel-reactive antibodies (of >80%) and no potential living kidney donor, or candidates who had (a) donor-specific HLA antibody(ies) (DSA) against their future donor (in cases with a living donor) and had been referred to our center for HLA desensitization before kidney transplantation (KT). Following the ENGAGE guidelines [10,11], all patients receiving a kidney from a living donor started from category 1 (cross-match positive) before desensitization and reached category 2–3 (lymphocytotoxicity [LCT] cross-match negative, historical DSA), while patients receiving a kidney from a deceased donor started from category 1–3 (DSA positive before desensitization) and reached category 2–3.

Patients were included between May 2016 and February of 2020. This study involving human participants was approved by CNIL [French national committee for data privacy], approval number 1987785v0. The patients provided their written informed consent to participate in this study. This work was performed following the principles of the Declaration of Helsinki (for patients’ protection) and Istanbul (for ethical organ procurement). No potentially identifiable human images or data are presented in this study. The raw data supporting the conclusions of this article will be made available by the authors, without undue reservation.

The desensitization strategy was either our SoC strategy or a tocilizumab + SoC (Toci + SoC) strategy. Both schemes are summarized in Figure 1, for living donor (1A) or deceased donor (1B) kidney transplantation. The SoC strategy began with infusion of rituximab (375 mg/m^2^), followed by a second infusion either 2 weeks after the first (in a deceased donor kidney setting), or at 10 days prior to living donor KT. Maintenance immunosuppression was started at the beginning of the process (deceased donor) or at the second rituximab infusion (living donor), using tacrolimus (0.1 mg/kg/day, to target trough levels of ~8–12 ng/mL), plus mycophenolate mofetil (500 mg b.i.d.) and prednisone (10 mg/day).

Apheresis was performed using semi-specific immuno-adsorption (IA) with GLOBAFFIN^®^ columns, starting either 5 days after the first rituximab infusion for deceased donor KTs or around 2 weeks before transplantation for living donor KTs (owing to the scheduled nature of the KT). The number of apheresis sessions was decided upon by the attending physician and adapted to the decrease of antibody MFI, i.e., targeting MFI values of <3000. In the tocilizumab + SoC (Toci + SoC) group, patients received a monthly infusion of tocilizumab (8 mg/kg) over 6 months and then received SoC.

The choice between SoC and Toci + SoC was based upon initial mean fluorescent intensities (MFIs) as it became apparent that elevated MFI antibodies (>15,000) tended to be more difficult to deplete. Thus, the two groups were not comparable with respect to the sensitization intensity, in terms of MFIs. Consequently, we later adjusted our comparisons on baseline MFI values.

Kidney transplantations were performed only when there was a negative LCT crossmatch on the last available serum at pre-transplantation. HLA-C or HLA-DP DSA could be detected at the time of transplantation with a MFI > 3000, since these C and DP loci are not exclusion criteria for a kidney offer. Maintenance immunosuppression after transplantation was based on tacrolimus (trough concentrations 8–12 g/mL for the first month, then 5–8 ng/mL), mycophenolate mofetil (1 g bid over the first two weeks, then 500 mg bid) and steroids (500 mg IV bolus of methylprednisolone, tapered down to 10 mg/day over the first week, then maintained to 5–10 mg/day at least until 1-year post-transplantation).

We evaluated kidney function through estimated glomerular filtration rate (eGFR) at 1-year post-KT, using the CKD-EPI estimator.

### 2.2. Evaluation of HLA Antibodies

In this high-risk immunological setting, the HLA antibodies were evaluated using single-antigen bead assays (Immucor, Inc., Norcross, GA, USA) on a Luminex platform for all serum samples. Raw MFI values on neat sera were provided for each HLA allele available in the kit panel, allowing tracking of MFI evolution of each antibody over time, both before and after transplantation. Sera with high MFIs (>15,000) were diluted (1/5 and 1/10) to check for a prozone effect [12]. Only the beads with an MFI > 1000 were considered in this analysis. We focused our analysis on three specific time-points: pre-desensitization (baseline), on the day of KT (D0), and at 1-year post-transplantation (Y1). We defined the pre-KT delta MFI and, similarly, the post-KT delta MFI, either as an absolute value (MFIbaseline–MFID0 and MFIY1–MFID0) or a reduction ratio ([MFIbaseline–MFID0]/MFIbaseline and [MFIY1–MFID0]/MFID0). These ratios were used to account for the semi-quantitative nature of MFIs (therefore, a reduction of 1000 MFI units is not perceived equally for a baseline MFI of 3000 or 15,000). MFIs are presented as medians with interquartile ranges [1Q–3Q].

To approach the clinical reality based on anti-HLA antibody presence/absence and not MFIs, we also counted the number of significant positive antibodies using a 3000-MFI threshold to define clinically significant positivity. We defined an eliminated antibody as one for which there was a decrease from a MFI > 3000 to a MFI < 3000 for a given antibody, provided there was a minimal 1000 MFI unit reduction (to avoid counting, as eliminated, an antibody with a small MFI reduction around an initial value of 3000). We used the same threshold to define acceptable incompatibilities for patients on the waiting list (any antibody with a lower MFI was deemed acceptable). This 3000 MFI threshold was chosen, as it correlates well with a negative flow-cytometry crossmatch.

Calculated panel reactive antibodies (cPRA) scores were computed using the online US Organ Procurement and Transplantation Network (OPTN) cPRA calculator (https://optn.transplant.hrsa.gov/data/allocation-calculators/cpra-calculator/ (accessed on 1 April 2022)). Three different MFI thresholds were compared to define the unacceptable antigens: MFI 500, 1500, and 3000.

We finally performed the same analyses focusing on donor-specific HLA antibodies (DSAs). These DSAs could either be defined in advance (for living donors) or defined a posteriori (for deceased donors, whose HLA typing was obviously unknown prior to transplantation).

### 2.3. Statistical Analyses

To assess the impact of the two treatment strategies on HLA antibodies, we compared the SoC and Toci + SoC groups using simple univariate tests, but also multivariate models. A first series of models was used to predict pre-KT delta MFI, adjusting for MFI baseline, the desensitization groups, and the HLA classes. We used a fixed-effect model and a mixed-effect model, adjusting individual patients as a random covariate, with an interaction term between MFI baseline and the desensitization groups. This interaction term accounted for a potential differential effect of the desensitization group, depending on MFI baseline, i.e., the desensitization strategy would induce a different MFI reduction depending on the initial MFI.

A second set of models was built to predict the post-Tx delta MFI, based on the pre-Tx delta MFI, i.e., how the potential benefit of desensitization persisted over the first year post-transplantation. We again adjusted for treatment group (SoC vs. Toci + SoC), HLA class, and used an interaction term between pre-KT delta and desensitization groups. This interaction term accounted for the potential association between the desensitization strategy efficacy (i.e., pre-Tx delta MFI) and the post-KT MFI rebound.

In general, the multivariate models allowed us to differentiate the effect of tocilizumab and the initial aforementioned indication bias (we had higher baseline MFI in the tocilizumab group).

## 3. Results

### 3.1. Description of the Patients’ Population

Between 2016 and 2020, 15 patients received tocilizumab as a desensitization therapy. Among these, seven could finally receive a kidney transplant following Toci + SoC desensitization. Reasons for not proceeding to transplantation were intercurrent events (infectious or cardiovascular, *n* = 4), a specific desensitization protocol different from SoC (*n* = 2), and patient refusal or non-compliance (*n* = 2).

A total of 33 patients were included in our analysis: 26 patients received the SoC treatment (SoC group) and 7 patients received Toci + SoC. The demographic characteristics of the two groups are detailed in Table 1. The median value of panel-reactive alloantibodies (PRA) was 96%, but three patients had a PRA lower than 50% (all being living kidney donor candidates with a DSA against their donor).

Initial median individual MFIs were different between treatment groups (3862 [IQ 1636–8116] vs. 4792 [IQ 2203–11145] for class 1 antibodies, *p* < 0.001; 3247 [IQ 1590–9188] vs. 6160 [IQ 2412–11603] for class 2 antibodies, *p* < 0.001, SoC vs. Toci + SoC). Owing to the indication bias of Toci + SoC, HLA sensitization before desensitization was broader in the Toci + SoC group. The initial median numbers of antibodies with a MFI > 3000 were 19 and 29 for class 1 in the SoC and Toci + SoC groups, respectively (overall median MFI: 4575 and 4662, respectively), and 5 and 19 for class 2 antibodies (overall median MFI: 2608 and 9406, respectively).

The median time to transplantation, from the beginning of desensitization, was 29 (IQ 27–43) days in the SoC group and 252 (IQ 242–285) in the Toci + SoC group. Specifically for deceased donors, the median time to transplantation was 49 (IQ 35–59.5) in the Soc group and 252 (IQ 246–266) in the Toci + SoC group.

In this study, we only included tocilizumab-treated patients that could achieve transplantation. However, there was only one tocilizumab-treated patient who could not be transplanted, due to an abscess of his arterio-veinous fistula, positive for methicillin-sensitive Staphylococcus aureus. This patient was not included in the analysis because it was not possible to follow his post-transplantation HLA sensitization. Further tolerance data for our whole desensitized cohort, including the 33 patients considered here, were previously published [13] and we refer the interested reader to this analysis.

### 3.2. Effect of Tocilizumab in the Desensitization Process (Pre-Transplantation)

We first compared the overall effect of the two desensitization strategies in terms of individual MFIs for all antigens. In univariate analysis, there was a difference in MFI reduction between both groups (as shown in Figure 2A for absolute delta MFIs and Figure 2B for relative delta MFIs). However, this difference was mainly driven by the initial differences between groups: in multivariate analysis, pre-Tx delta MFIs were associated with the MFI baseline (coefficient 0.78, *p* < 0.001) but not with the treatment group (*p* = 0.87).

Given the limits of this median MFI interpretation, we also investigated the number of eliminated antibodies, whose MFI became <3000 over the desensitization process amongst antibodies with an initial MFI > 3000 (as shown in Figure 2C). There was a more intense antibody elimination (with a 3000-MFI threshold) in favor of Toci + SoC for class 1 (*p* = 0.045), but a non-significant trend in favor of Toci + SoC for class 2 antibodies (*p* = 0.11).

When the initial differences in HLA antibodies between the two groups were accounted for, there was no tocilizumab effect. Indeed, in the multivariate analysis, a high initial number of antibodies with MFI of >3000 was associated with increased desensitization efficiency (*p* < 0.001), but neither the desensitization treatment nor the HLA class were associated with antibody elimination.

When considering calculated panel-reactive antibodies (cPRA), we confirmed a clear and significant cPRA reduction over the desensitization process (Figure 3A), regardless of the cPRA definition. Additionally, there was no difference between treatment groups in terms of cPRA reduction (Figure 3B), regardless of the MFI threshold for inclusion of an antibody in the analysis.

### 3.3. One-Year Post-Transplantation Antibody Reconstitution

After 1-year post-transplantation, compared to day 0, the absolute MFI increase was negligible, with a median of +255 (with a very wide range, from −17,941 to 18,123). When investigating HLA class and treatment group, the median absolute MFI increase (from D0 to Y1) was +222 [−192–+1216] and +55 [−526–+357] in class 1 for SoC and Toci + SoC (*p* < 0.001), respectively, and +1013 [−392–+4217] and +520 [+32–+1542] in class 2 (*p* = 0.327).

However, there was an association between the decrease in pre-transplant MFI and the post-transplant increase (Figure 4). Interestingly, the Toci + SoC strategy significantly limited the increase in those antibodies that had the greatest pre-transplantation decrease, i.e., the strongest antibodies (interaction between pre-Tx delta MFI and treatment group, multivariate model, *p* < 0.001). In other words, the use of tocilizumab specifically limited the rebound of high-MFI antibodies. When investigating this effect within each HLA subclass, we found that this tocilizumab effect was true only for class 2 antibodies (*p* < 0.001 for interaction between pre-Tx delta MFI and treatment group). Using the mixed-effect model for intra-patient variability, similar results were obtained.

We also focused on recurring antibodies (between D0 and 1-year, with a MFI threshold > 3000) to evaluate allo-reconstitution 1-year post-transplantation. Most patients did not develop historical or de novo HLA antibodies with a MFI > 3000 post-transplantation (median number of recurring antibodies: 0 [0–12] for class 1, 0 [0–34] for class 2). There was no difference between treatment groups regarding a post-transplantation increase in positive antibodies in univariate (*p* = ns) or multivariate analysis (*p* = ns, adjusting for baseline MFI, number of eliminated antibodies pre-transplantation, and HLA class).

### 3.4. DSA Evaluation and Focus on Patients with DSA Increase

While considering all anti-HLA antibodies gives a perspective on overall immune reconstitution after desensitization with or without tocilizumab, focusing on DSA antibodies allowed us to evaluate the added value of tocilizumab on the specific immunologic risk for a given transplant.

Overall, there were initially a median of 1 [IQ 1–2] historical DSAs for the SoC group compared to 3 [IQ 1–3.75] for the Toci + SoC group. Figure 5 describes the individual DSA MFIs in detail and shows their evolution over time, by treatment and HLA class. When available, data from 1-month and 3-months post-transplantation are shown.

We compared the increase of DSA MFIs between D0 (time of transplantation) and 1-year post-transplantation, as well as DSA recurrence (with a 3000-MFI threshold) over the same period. In univariate analysis, there was no difference between the treatment groups, with a similar DSA MFI rebound after transplantation (*p* = ns in both classes). Additionally, in multivariate analysis, and adjusting for the initially different DSA MFIs, there was no effect of tocilizumab on post-transplantation DSA MFIs (regression coefficient −223.4, *p* = 0.71). In a similar multivariate model, the number of recurring DSAs at post-transplantation (with the same 3000 MFI threshold) did not differ between the treatment groups (*p* = 0.87). Finally, predicting the probability of individual DSA recurrence after transplantation, based on each DSA baseline MFI and treatment group, there was no effect of tocilizumab either (logistic regression, *p* = ns).

A total of 4 patients had a DSA with a post-transplantation increase.

Patient 3 (SoC treatment) underwent a living donor kidney transplantation. There was a weak baseline DSA against DQB1*03:02 with a MFI of 2191. This DSA was not detected on the day of transplantation (MFI 357) but culminated at 6199 at 1-year post-transplantation. Over the following 17 available samples, up to 5-years post-transplantation, the mean MFI for this DSA was 2036 ± 1530. There was no BPAR in the first year.

Patient 4 (SoC treatment) underwent a living donor kidney transplantation. There was a serologic pre-transplantation DQ6 antibody that was not allele specific (DQB1*06:01 with a MFI of 11,787 at baseline, DQB1*06:02 (specific target) with a MFI of 546 at baseline). There was no BPAR in the first year.

Patient 9 (SoC treatment) underwent a deceased donor kidney transplantation, with a DPB1*02:01 DSA. This DSA MFI was 6191 at baseline, with a mean MFI of 8515 ± 1721 over the 10 available samples pre-transplantation. This patient underwent a BPAR episode in the first year.

Patient 22 (Toci + SoC treatment) underwent a deceased donor kidney transplantation, with a DSA against DPB1*03:01 and 6 other class 1 and class 2 DSA. Only the DPB1*03:01 recurred with 1-year post-transplantation MFI at 4631. It had the highest baseline MFI, at 13,242. Further MFI at 2- and 3-years post-transplantation were of the same magnitude, at 3767 and 6745. There was no BPAR in the first year.

### 3.5. One-Year Outcomes of Transplanted Kidneys

In terms of 1-year graft survival, there were two graft losses in the SoC group (day 5 and day 27) and none in the Toci + SoC group. Death-censored graft survival did not differ between the treatment groups (log-rank test: *p* = 0.46). The median eGFR at 1-year post-transplantation was 62.8 [53.3–78.7] in the SoC group and 65.6 [59.3–79.5] mL/min/1.73 m^2^ in the Toci + SoC group (*p* = ns). Individual eGFR trajectories are presented in Appendix A.

### 3.6. Kidney Histology over the First-Year Post-Transplantation

Protocol biopsies were performed at 1-, 3-, and 12-months post-transplantation, when possible. Over the first-year post-transplantation, 31/33 patients had at least one allograft biopsy, including for-cause biopsies, for a total of 78 biopsies. The proportions of any diagnosis of ABMR within 1-year post-transplantation was 25% in the SoC group vs. 57.1% in the Toci + Soc group (difference of proportions: *p* = 0.254). Isolated C4d lesions were identified on 12/18 (40%) of the biopsies. Individual Banff items associated with ABMR are presented in Figure 6.

## 4. Discussion

In this study, we investigated the differential impact of two different desensitization strategies, namely tocilizumab + SoC, or SoC alone (rituximab + apheresis). We have shown that using tocilizumab as an add-on therapy induced a more intense relative reduction of MFI, however not clinically significant. There was a tendency towards a greater reduction in the number of positive antibodies in the Toci + SoC group, with a 3000-MFI threshold. Tocilizumab also induced a significantly lower post-transplantation rebound of high MFI antibodies. In other words, tocilizumab seems to induce a longer-lasting effect of desensitization on high MFI antibodies post-transplantation, thus limiting their rebound.

Access to transplantation may be improved using tocilizumab together with apheresis, rituximab, and IVIg. Our data suggest a significant impact of tocilizumab on humoral memory and its potential use in desensitization [14,15].

Our study has several limitations due to its small sample size (*n* = 33) and its uncontrolled design in a fast-moving field that requires treatment adaptations over time. Our SoC and Toci + SoC groups differed, especially in terms of class-1 antibody MFIs, but accounting for this difference in our rebound prediction left our message unchanged: the use of tocilizumab may help temper the humoral reaction post-transplantation and so deserves further investigation. Larger studies are highly needed to confirm or question our results.

Interleukin-6 inhibitors have been recently used in cases of desensitization and to treat antibody-mediated rejection, after the seminal study by Vo et al. [14]. As suggested in this study, the continued use of tocilizumab after transplantation might be an interesting prophylactic treatment for ABMR after transplantation. In our cohort, there was a nonsignificant tendency toward more BPAR in the Toci + SoC group, which might be explained by the more intense DSAs in this group. This BPAR risk might be mitigated by maintaining monthly tocilizumab injections after transplantation.

More recently, tocilizumab has been used to treat ABMR but showed no benefit at 1-year post-treatment in a cohort of 9 ABMR patients compared to a matched historical cohort of 37 patients [16]. However, in a phase II randomized controlled trial, clazakizumab, a monoclonal anti-Il-6 antibody, had a significant impact on ABMR, although infections were more serious in the clazakizumab group [17].

Mechanistic studies can help us decipher these conflicting results. The effect of tocilizumab in the setting of subclinical graft inflammation was recently investigated in kidney transplant recipients [18]. Using surveillance allograft biopsies performed during the first-year post-transplantation to detect subclinical graft inflammation, the balance between T regulatory and effector functions was shown to be biased toward a more pro-tolerant response when using tocilizumab compared to controls (receiving tacrolimus-based immunosuppression). In this study, there were significant decreases in the frequencies of IFN-gamma-producing T CD4 and T CD8 cells (after ex vivo polyclonal stimulation) among tocilizumab-treated patients compared to controls (*p* = 0.0208 in both cases) over the 6-month follow-up.

In a different clinical setting, we also investigated the effect of tocilizumab on the immune response of the same tocilizumab-treated cohort of patients described here [9]. We identified a defect in B cell maturation, with a decrease in all post-germinal center B subpopulations during the treatment period. In our cohort, the Tregs did not evolve over the 6-month treatment period.

Overall, two main broad mechanistic explanations can be devised to explain the efficacy of tocilizumab in preventing the antibody rebound: either a non-specific anti-inflammatory effect, as demonstrated by the limit in CRP increase in tocilizumab-treated patients, or a specific control of antibody-producing cell precursors. The various possible effects of tocilizumab were recently reviewed [19,20].

The results of this study, in line with these recent investigations, suggest the importance of the interleukin-6-dependent pathway to maintain humoral allo-reactivity. Further mechanistic trials are needed to elucidate the effects of tocilizumab. Also, randomized clinical trials are expected to evaluate the pros and cons of IL-6 blocking. Owing to the clinical burden of alloreactivity in solid-organ transplantation, this is a major challenge, yet a major way forward for our patients.

## Figures and Tables

**Figure 1 jcm-12-00424-f001:**
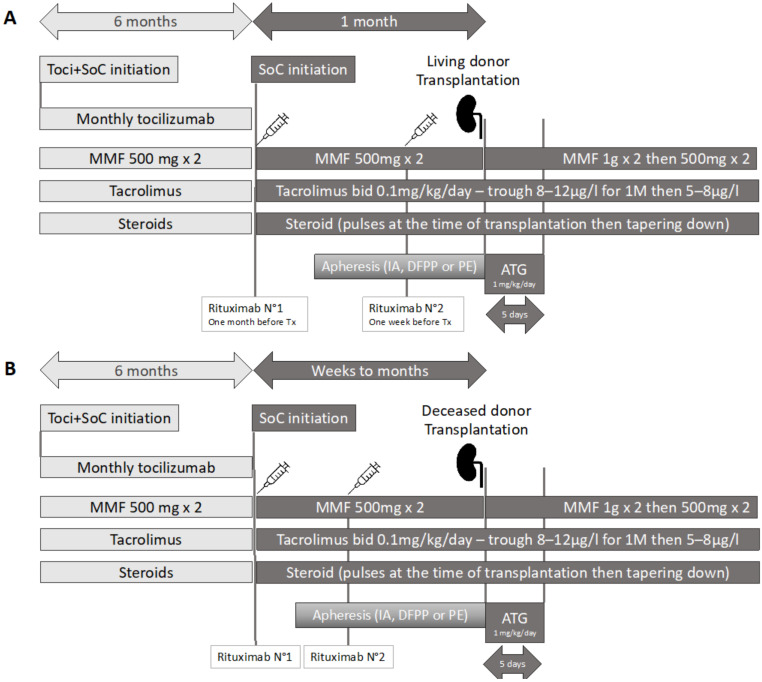
Overview of the desensitization process, (**A**) for the living donor kidney transplantation, (**B**) for deceased donor kidney transplantation. All patients followed the dark gray path; only Toci + SoC patients followed the light gray path before receiving the SoC.

**Figure 2 jcm-12-00424-f002:**
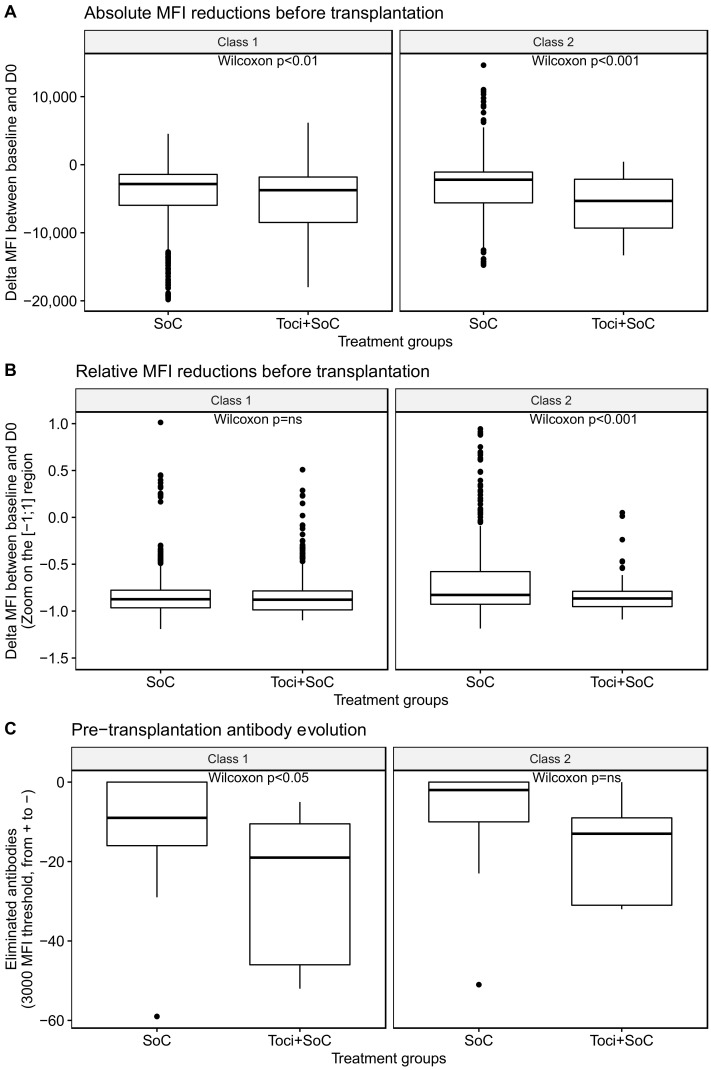
Evolution of MFI and positive antibodies (MFI threshold 3000) over the desensitization process (i.e., before transplantation). (**A**) Absolute MFI reduction per antibody. (**B**) Relative MFI reduction per antibody, with respect to the baseline MFI. (**C**) Number of antibodies for which MFI was higher than 3000 and was reduced to less than 3000, with a minimum absolute delta of 1000. SoC group: 26 patients; Toci + SoC group: 7 patients.

**Figure 3 jcm-12-00424-f003:**
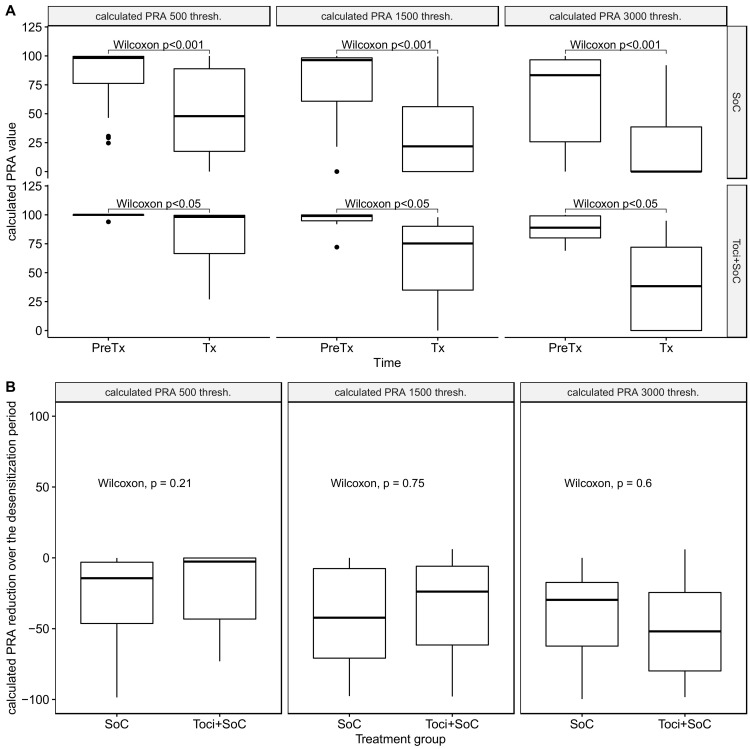
Evolution of calculated panel-reactive antibodies (cPRA), with different MFI threshold for antibody positivity. (**A**) Evolution of cPRA through the desensitization process, by treatment group and by cPRA definition. (**B**) Comparison of cPRA reduction by treatment group, by cPRA definition. SoC group: 26 patients; Toci + SoC group: 7 patients.

**Figure 4 jcm-12-00424-f004:**
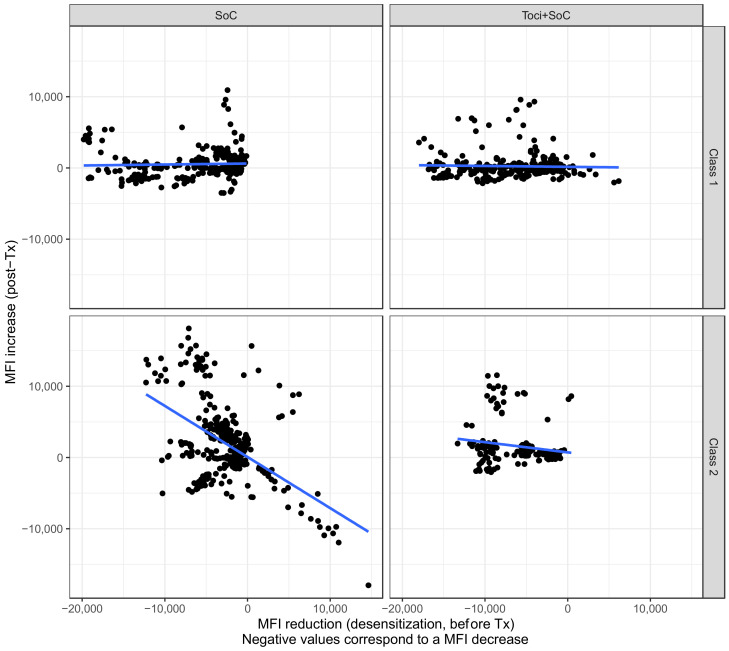
Comparison of pre-transplantation absolute MFI reduction (over the course of desensitization) and post-transplantation rebound for the two desensitization strategies and the two HLA classes. Blue lines are linear regression lines.

**Figure 5 jcm-12-00424-f005:**
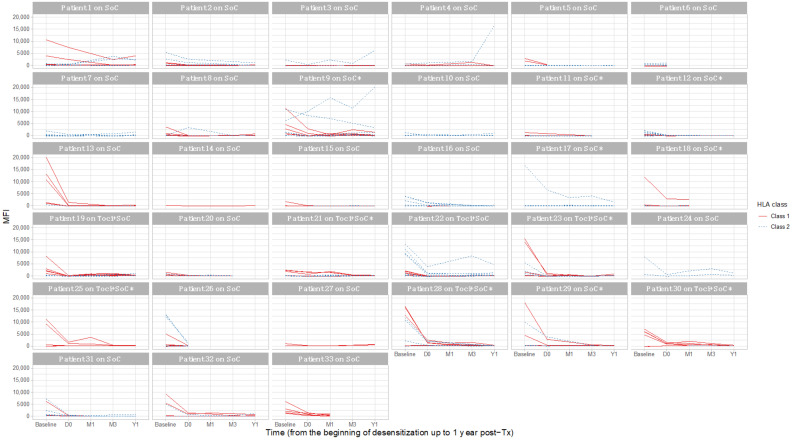
Evolution of donor-specific antibodies after transplantation, depending on the desensitization treatment. Toci + SoC: Tocilizumab + standard of care; SoC: standard of care only. A star indicates a BPAR episode in the 1st year post-transplantation.

**Figure 6 jcm-12-00424-f006:**
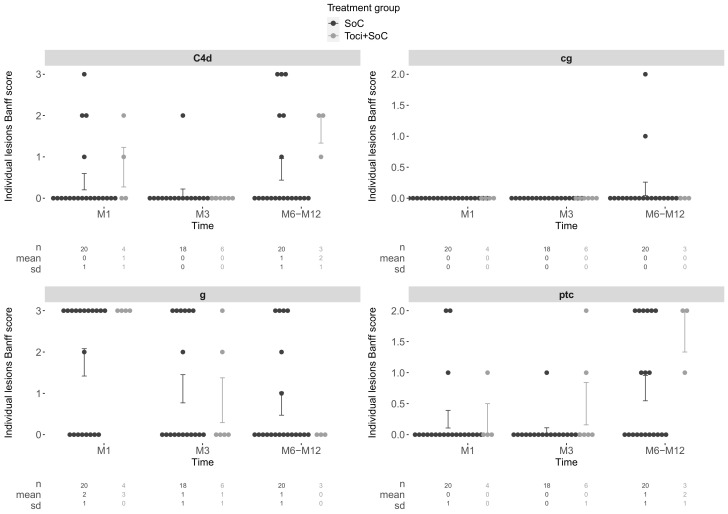
Individual Banff score items (C4d, cg, g, and ptc) over time for all systematic biopsies performed at M1 and M3 and between M6 and M12 post-transplantation. The two treatment groups are represented in black (SoC group, 26 patients) and grey (Toci + Soc group, 7 patients). Mean and standard-error bars are represented.

**Table 1 jcm-12-00424-t001:** Demographic characteristics of the study population.

	SoC (N = 26)	Toci + SoC (N = 7)	Total (N = 33)	*p*
Recipient’s age, yr				0.86
Median (Q1, Q3)	55.10 (42.78, 61.72)	50.50 (45.20, 56.90)	51.40 (42.40, 61.80)	
Recipient’s gender (male)	11 (42%)	3 (43%)	14 (42%)	0.98
Donor type				0.03
DCD	0 (0%)	1 (14%)	1 (3%)	
BD	11 (42%)	5 (71%)	16 (48%)	
LD	15 (58%)	1 (14%)	16 (48%)	
Dialysis vintage, yr				0.12
Median (Q1, Q3)	3.32 (1.55, 8.58)	8.86 (7.52, 17.88)	5.36 (1.69, 9.33)	
Historical PRA, %	95.00 (79.75, 99.00)	98.00 (96.50, 99.50)	96.00 (85.00, 99.00)	0.19
Number of historical DSA Median [min–max]	2 (1–7)	5 (2–7)	3 (1–7)	0.01
Highest MFI DSA before desensitization (median, IQR)				0.06
Class 1	5592 (1663–10,794)	8185 (4719–13,328)	5016 (1663–10,132)	
Class 2	4442 (1718–8451)	11,672 (8590–12,457)	9694 (5897–12,850)	
Month-3 post-Tx GFR, mL/min (median, IQR)	60.32 (53.95, 89.23)	75.05 (42.10, 75.39)	61.30 (51.15, 86.94)	0.61
Year-1 post-Tx GFR, mL/min (median, IQR)	62.81 (53.33, 78.66)	65.64 (59.31, 79.50)	64.22 (53.24, 79.57)	0.87

Abbreviations: SoC, standard of care; toci, tocilizumab; Q1, Q3: 1st and 3rd quartiles; DCD, donor after cardiac death; BD, heart-beating donor; LD, living donor; yr, years; DSA: donor-specific antibody; MFI: mean fluorescent intensity; PRA, panel-reactive alloantibodies.

## Data Availability

The data presented in this study are available on request from the corresponding author.

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
