# Peer review of "Tocilizumab Evaluation in HLA-Desensitization before Kidney Transplantation as an Add-On Therapy to Apheresis: The TETRA Study"

_jcm, 2023, doi:10.3390/jcm12020424_

Round 1
Reviewer 1 Report
overall interesting and important manuscribt describing the effect of Tocilizumab in 7 patients undergoing a HLA desensitization protocol.
Comments:
The description of the desensitization protocol is somewhat difficult to follow and the manuscript would benefit from a separate figure (1b) for deceased donor desensitization, especially as most patients received a deceased donor transplant.
The ENGAGE group (Bestard O et al Transplant Int. 2021 and 2022) recently defined 5 immunological risk categories, which should be included in the description of the patient cohort incl Table 1
The authors should also describe immunosuppressive therapy after transplantation in greater detail, e.g. Tacrolimus levels, MPA doses, and steroid on week 1, month 1, 3, 6, and 12 as well as ATG and Rituximab doses and number of plasmapgheresis sessions in a supplementary table
what is the difference between a microlymphocytotoxicity test and a regular LCT.
regarding decreasing MFI values: elimination was defined as MFI<3000, but shouldn´t it also include a significant percentage change (e.g. a HLA Antibody would count as "eliminated" if MFI decreases fromm 3001 to 2999)?
The interpretation of the multivariate model with only n=33 patients and only n=7 Tocilizumab treated patients for me is difficult, especially given the strong indication bias and should tune down their interpretation
The authors should present a figure with the patient flow, as it is important to also visualize those patients, who interrupted Tocilizumab due to complications
I am puzzled by the fact that in Figure 2 high p-values are noted although boxplots widely overlap and only n=7 patients received Tocilizumab and finally non-significant differences in models and text: correct statistics?? correct p-values? re design figures?
figure 3: please add the number of patients in the graphs
Table 1 should include number of patients with pre transplant DSA and their characterisitcs (class I? II?, MFI?)
Results and Discussion: should tune down the positive effects on Tocilizumab (low numbers, indication bias, limited interpretation of statistics)
Author Response
Please find our response in the associated file.

Reviewer 2 Report
I appreciate to have a chance to review this interesting manuscript. This study referred to the important issues in kidney transplantation; desensitization of preformed DSA comparing SoC and tocilizumab+SoC. HOwever, I have some questions.
1. The reason for the cutoff of MFI < 3000 for the acceptatble decrease should be mentioned.
2. This study focused on the impact of these 2 protocols on MFI levels and rejection. The adverse events should be mentioned.
3. The impact of these 2 regimens on the non-DSA production after KT is our great concern,
Author Response
Please find our response in the attached file.

Reviewer 3 Report
In this single-center non-randomized prospective study the authors analysed tocilizumab on top of their standard of care regimen (consisting of rituximab and immunoadsorption with GLOBAFFIN columns) for desensitization. In an attempt to treat especially those patients with the highest PRA (and thus with the highest risk) with a potentially effective agent for desensitization they refrained from a clear group design. To compensate for this disadvantage, a sophisticated multivariate model, including treatment as a separate variable, was used in the statistics.
I have only minor comments/questions:
1.) What is CNIL in section 2.1? Is this equivalent to an IRB or are they only interested in data protection rather than patients' other rights? If you have approval from an ethics committee or have followed the principles of the Declaration of Helsinki and Istanbul, a short addendum indicating this would be appropriate.
2.) section 2.2 line 99-100: a description regarding sample preparation is lacking; is the complement interference problem not so big with immucore beads? (Schnaidt et al. Transplantation 2011, Schwaiger et al. Transplantation 2014, Greenshields and Liwski Hum Immunol 2019)
3.) section 3.4 line 265 ff : why was patient 9 transplanted at all? It seems as if his DPB1*02:01 DSA was never below 3000 MFI?
4.) section 4 line 322: maybe better explain here what clazakizumab is (i.e. an anti-IL-6 antibody). I think it was not mentioned before
Author Response

(The authors gave the same response as above.)
